# Autonomous Learning of Object-Centric Abstractions for High-Level Planning

## Abstract

We propose a method for autonomously learning an object-centric representation of a continuous and high-dimensional environment that is suitable for planning. Such representations can immediately be transferred between tasks that share the same types of objects, resulting in agents that require fewer samples to learn a model of a new task. We first demonstrate our approach on a simple domain where the agent learns a compact, lifted representation that generalises across objects. We then apply it to a series of Minecraft tasks to learn object-centric representations, including object types—directly from pixel data—that can be leveraged to solve new tasks quickly. The resulting learned representations enable the use of a task-level planner, resulting in an agent capable of forming complex, long-term plans with considerably fewer environment interactions.[1]

## 1 Introduction

Model-based methods are a promising approach to improving sample efficiency in reinforcement learning. However, they require the agent to either learn a highly detailed model—which is infeasible for sufficiently complex problems (Ho et al., 2019)—or to build a compact, high-level model that abstracts away unimportant details while retaining only the information required to plan. This raises the question of how best to build such an abstract model. Fortunately, recent work has shown how to learn an abstraction of a task that is provably suitable for planning with a given set of skills (Konidaris et al., 2018). However, these representations are highly task-specific and must be relearned for any new task, or even any small change to an existing task. This makes them fatally impractical, especially for agents that must solve multiple complex tasks.

We extend these methods by incorporating additional structure—namely, that the world consists of objects, and that similar objects are common amongst tasks. This can substantially improve learning efficiency, because an object-centric model can be reused wherever that same object appears (within the same task, or across different tasks) and can also be generalised across objects that behave similarly—object *types*. We assume that the agent is able to individuate the objects in its environment, and propose a framework for building portable object-centric abstractions given only the data collected by executing high-level skills. These abstractions specify both the abstract object attributes that support high-level planning, and an object-relative lifted transition model that can be instantiated in a new task. This reduces the number of samples required to learn a new task by allowing the agent to avoid relearning the dynamics of previously seen object types.

We make the following contributions: under the assumption that the agent can individuate objects in its environment, we develop a framework for building portable, object-centric abstractions, and for estimating object types, given only the data collected by executing high-level skills. We also show how to integrate problem-specific information to instantiate these representations in a new task. This reduces the samples required to learn a new task by allowing the agent to avoid relearning the dynamics of previously-seen objects.

We demonstrate our approach on a Blocks World domain, and then apply it to a series of Minecraft tasks where an agent autonomously learns an abstract representation of a high-dimensional task from raw pixel input. In particular, we use the probabilistic planning domain definition language (PPDDL) (Younes & Littman, 2004) to represent our learned abstraction, which allows for the use of existing

---

[1] More results and videos can be found at: `https://sites.google.com/view/mine-pddl`

task-level planners. Our results show that an agent can leverage these portable abstractions to learn a representation of new Minecraft tasks using a diminishing number of samples, allowing it to quickly construct plans consisting of hundreds of low-level actions.

## 2 BACKGROUND

We assume that tasks are modelled as semi-Markov decision processes $\mathcal{M} = \langle \mathcal{S}, \mathcal{O}, \mathcal{T}, \mathcal{R} \rangle$ where (i) $\mathcal{S}$ is the state space; (ii) $\mathcal{O}(s)$ is the set of temporally-extended actions known as *options* available at state $s$; (iii) $\mathcal{T}$ describes the transition dynamics, specifying the probability of arriving in state $s'$ after option $o$ is executed from $s$; and (iv) $\mathcal{R}$ specifies the reward for reaching state $s'$ after executing option $o$ in state $s$. An option $o$ is defined by the tuple $\langle I_o, \pi_o; \beta_o \rangle$, where $I_o$ is the *initiation set* that specifies the states in which the option can be executed, $\pi_o$ is the *option policy* which specifies the action to execute, and $\beta_o$ specifies the probability of the option terminating execution in each state (Sutton et al., 1999).

We adopt the object-centric formulation from Ugur & Piater (2015): in a task with $n$ objects, the state is represented by the set $\{\mathbf{f}_a, \mathbf{f}_1, \mathbf{f}_2, \ldots, \mathbf{f}_n\}$, where $\mathbf{f}_a$ is a vector of the agent's features and $\mathbf{f}_i$ is a vector of features particular to object $i$. Note that the feature vector describing each object can itself be arbitrarily complex, such as an image or voxel grid—in this work we use pixels.

Our state space representation assumes that individual objects have already been factored into their constituent low-level attributes. Practically, this means that the agent is aware that the world consists of objects, but is unaware of what the objects are, or if there are multiple instantiations of the same object present. It is also easy to see that different tasks will likely have differing numbers of objects with potentially arbitrary ordering; any learned abstract representation should be agnostic to this.

### 2.1 STATE ABSTRACTIONS FOR PLANNING

We intend to learn an abstract representation suitable for planning. Prior work has shown that a sound and complete abstract representation must necessarily be able to estimate the set of initiating and terminating states for each option (Konidaris et al., 2018). In classical planning, this corresponds to the *precondition* and *effect* of each high-level action operator (McDermott et al., 1998).

The precondition is defined as $\text{Pre}(o) = \text{Pr}(s \in I_o)$, which is a probabilistic classifier that expresses the probability that option $o$ can be executed at state $s$. Similarly, the effect or *image* represents the distribution of states an agent may find itself in after executing $o$ from states drawn from distribution $Z$ (Konidaris et al., 2018): $\text{Im}(Z, o) = \frac{1}{G} \int_{\mathcal{S}} \text{Pr}(s' \mid s, o) Z(s) \text{Pr}(s \in I_o) ds$, where $G = \int_{\mathcal{S}} Z(s) \text{Pr}(s \in I_o)$. Since the precondition is a probabilistic classifier and the effect is a probabilistic density estimator, they can be learned directly from option execution data.

We can use preconditions and effects to evaluate the probability of a sequence of options—a plan—executing successfully. Given an initial state distribution, the precondition is used to evaluate the probability that the first option can execute, and the effects are used to determine the resulting state distribution. We can apply the same logic to the subsequent options to compute the probability of the entire plan executing successfully. It follows that these representations are sufficient for evaluating the probability of successfully executing *any* plan (Konidaris et al., 2018).

**Partitioned Options** For large or continuous state spaces, estimating $\text{Pr}(s' \mid s, o)$ is difficult because the worst case requires learning a distribution conditioned on every state. However, if we assume that terminating states are independent of starting states, we can make the simplification $\text{Pr}(s' \mid s, o) = \text{Pr}(s' \mid o)$. These *subgoal* options (Precup, 2000) are not overly restrictive, since they refer to options that drive an agent to some set of states with high reliability. Nonetheless, many options are not subgoal. It is often possible, however, to *partition* an option's initiation set into a finite number of subsets, so that it is approximately subgoal when executed from any of the individual subsets. That is, we partition an option $o$'s start states into finite regions $\mathcal{C}$ such that $\text{Pr}(s' \mid s, o, c) \approx \text{Pr}(s' \mid o, c), c \in \mathcal{C}$ (Konidaris et al., 2018). As in prior work (Andersen & Konidaris, 2017; Konidaris et al., 2018; Ames et al., 2018), we achieve this in practice by clustering options based on their terminating states.

**Factors** We adopt the frame assumption, which states that aspects of the world not explicitly affected by an agent's action remain the same (Pasula et al., 2004). Prior work leverages this to learn a factored or STRIPS-like (Fikes & Nilsson, 1971) representation by computing the option's *mask*: the state variables explicitly changed by the option (Konidaris et al., 2018). In our formulation, the state space is already factorised into objects, so computing the mask amounts to determining which objects are affected by a given option.

## 3 LEARNING PORTABLE OBJECT-CENTRIC REPRESENTATIONS

Although prior work (Konidaris et al., 2018) allows an agent to autonomously learn an abstract representation supporting fast task-level planning, that representation lacks generalisability—since the symbols are distributions over states in the current task, they cannot be reused in new ones. This approach can be fatally expensive in complex domains, where learning an abstract model may be as hard as solving a task from scratch, and is therefore pointless if we only want to solve a single task. However, an agent able to reuse aspects of its learned representation can amortise the cost of learning over many interactions, accelerating learning in later tasks. The key question is what forms of representation support transfer in this way.

One approach is to assume that the world consists of objects, and that similar objects are shared across tasks. Approaches like object-oriented MDPs (Diuk et al., 2008) exploit the presence of objects by providing the agent with an object-oriented representation, resulting in compact representations that are transferable between tasks sharing the same object classes and dynamics (Guestrin et al., 2003; Diuk et al., 2008; Marom & Rosman, 2018). Similarly, the classical planning literature has long represented problems in terms of the objects that constitute a domain, and operators that can affect their states (McDermott et al., 1998). In both cases, however, the question arises as to the most appropriate way of building an object-oriented representation of a problem, especially one experienced by the agent at the pixel level. This includes deciding which attributes should be chosen to characterise a particular type, as well as which objects should belong to each type.

We now introduce an object-centric generalisation of a learned symbolic representation that admits transfer in tasks when the state space representation consists of features centred on objects in the environment. This is common in robotics, where each object is often isolated from the environment and represented as a point cloud or subsequently a voxelised occupancy grid. We summarise our proposed approach in Figure 1 and the remainder of this section, and provide a detailed pseudocode description in Appendix F.

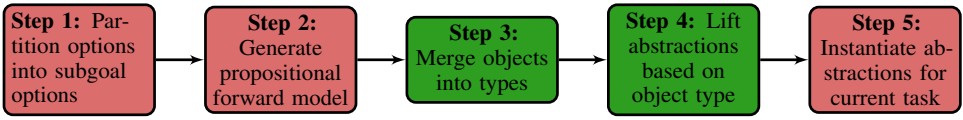

Figure 1: Learning lifted representations from data. Red nodes represent problem-specific representations, while green nodes are abstractions that can be transferred between tasks.

### 3.1 GENERATING A PROPOSITIONAL MODEL (STEPS 1–2) (AS IN KONIDARIS ET AL., 2018)

The agent begins by executing options using an exploration policy to collect transition data. The first step is to partition the options into approximately subgoal options. For each option $o$ and empirical sets of initial and terminating states $\tilde{I}_o$ and $\tilde{\beta}_o$, the agent partitions $I_o$ into subsets $K \subseteq \tilde{I}_o$ such that $\Pr(s' \mid s_i, o) = \Pr(s' \mid s_j, o) \forall s_i, s_j \in K, s' \in \tilde{\beta}_o$. In practice, this can be achieved by first clustering state transition samples based on terminating states, and then assigning each cluster to a partition. Finally, pairs of partitions whose initiating states overlap are merged to handle probabilistic effects.

The agent next learns a precondition classifier for each approximately partitioned option. Partitions' initiating states are used as positive examples, and all other states as negative ones. A feature selection procedure determines which objects are relevant to the precondition, and a classifier is fit using only those objects. A density estimator is then used to estimate the effect distribution for each partitioned

option. The agent learns distributions over only the objects affected by the option, learning one estimator per object. Together these state distributions form our propositional PPDDL vocabulary $\mathcal{V}$.

For each partitioned option $o$, the agent has learned a precondition classifier $\hat{I}_o$ and effect estimator $\hat{\beta}_o$. However, to construct a PPDDL representation, both the precondition and effects must be specified in terms of state distributions (propositions) only. Effects are modelled as such, and so pose no problem, but the learned precondition is a classifier rather than a state distribution. The agent must therefore iterate through all possible effect distributions to compute whether the skill can be executed there. This is achieved by replacing $o$'s precondition classifier with every $\mathcal{P} \in \wp(\mathcal{V})$ such that $\int_{\mathcal{S}} \hat{I}_o(s)\mathcal{G}(s)ds > 0, \mathcal{G} = \prod_{p \in \mathcal{P}} p$, where $\wp(\mathcal{V})$ denotes the powerset of $\mathcal{V}$. In other words, the agent considers every combination of effect state distributions and draws samples from their conjunction. If these samples are classified as positive by $\hat{I}_o$, then the conjunction $\mathcal{P}$ is used to represent the precondition. The preconditions and effects are now specified using distributions over state variables, where each distribution is a proposition. We have now learned a PPDDL representation, which is sound and suitable for planning.

## 3.2 Generating a Lifted, Typed Model (Steps 3–4)

At this point, the agent has learned an abstract, but task-specific, representation. Unfortunately, there is no opportunity for transfer (both within the task and between different tasks), because each object is treated as unique. To overcome this, we now propose a method for determining object *types*.

**Definition 1.** Assume that option $o$ has been partitioned into $n$ subgoal options $o(1), \ldots, o(n)$. Object $i$'s *profile* under option $o$ is denoted by

$$\text{Profile}(i, o) = \big\{ \{\text{Pre}_i^{o(1)}, \mathcal{E}_i^{o(1)}\}, \ldots, \{\text{Pre}_i^{o(n)}, \mathcal{E}_i^{o(n)}\} \big\},$$

where $\text{Pre}_i^{o(k)}$ is the distribution over object $i$'s states present in the precondition for partition $k$, and $\mathcal{E}_i^{o(k)}$ is object $i$'s effect distribution.[2]

**Definition 2.** Two objects $i$ and $j$ are *option-equivalent* if, for a given option $o$, $\text{Profile}(i, o) = \text{Profile}(j, o)$. Furthermore, two objects are *equivalent* if they are option-equivalent for every $o$ in $\mathcal{O}$.

The above definition implies that objects are equivalent if one object can be substituted for another while preserving every operator's abstract preconditions and effects. Such objects can be grouped into the same *object type*, since they are functionally indistinguishable for the purposes of planning. In practice, however, we can use a weaker condition to construct object types. Since an object-centric skill will usually modify only the object being acted upon, and because we have subgoal options that do not depend on the initial state, we can take a similar approach to Ugur & Piater (2015) and group objects by effects only:

**Definition 3.** Assume that option $o$ has been partitioned into $n$ subgoal options. Object $i$'s *effect profile* under option $o$ is denoted by

$$\text{EffectProfile}(i, o) = \big\{ \mathcal{E}_i^{o(1)}, \ldots, \mathcal{E}_i^{o(n)} \big\},$$

where $\mathcal{E}_i^{o(k)}$ is object $i$'s effect distribution. Two objects $i$ and $j$ are *effect-equivalent* if $\text{EffectProfile}(i, o) = \text{EffectProfile}(j, o)$ for every $o$ in $\mathcal{O}$.

By computing effect profiles using the propositional representation, the agent can determine whether objects $i$ and $j$ are similar (using an appropriate measure of distribution similarity) and, if so, merge them into the same object type. Propositions representing distributions over individual objects can now be replaced with predicates that are parameterised by types. For example, if there are four doors in a domain, then the agent can replace four propositions representing each door closed with a single `ClosedDoor` predicate parameterised by an object of type `door`.

## 3.3 Problem-Specific Instantiation (Step 5)

If the task dynamics are completely described by the state of each object, as is the case in object-oriented MDPs (Diuk et al., 2008), then our typed representation is sufficient for planning. However,

---

[2] These precondition and effect distributions can be null where appropriate.

in many domains the object-centric state space is *not* Markov. For example, in a task where only a particular key opens a particular door, the state of the objects alone is insufficient to describe dynamics—the identities of the key and door are necessary too. A common strategy in this case is to augment an ego- or object-centric state space with problem-specific, allocentric information to preserve the Markov property (Konidaris et al., 2012; James et al., 2020). We denote $\mathcal{X}$ as the space of problem-specific state variables. $\mathcal{S}$ remains the original object-centric state space. The above complication does not negate the benefit of learning transferable abstract representations, as our existing operators learned in $\mathcal{S}$ can be augmented with propositions over $\mathcal{X}$ on a per-task basis. In general, local information relative to individual objects will transfer between tasks, but problem-specific information, such as an object's global location, must be relearned each time.

For each partitioned option $o$ with sets of start and end states $I_o, \beta_o \subseteq \mathcal{S} \times \mathcal{X}$, the agent re-partitions $I_o$ such that $\Pr(x' \mid x_i, o) = \Pr(x' \mid x_j, o) \forall x_i, x_j \in \kappa, (\cdot, x') \in \beta_o$ for $\kappa \subseteq I_o$. This forms partitioned subgoal options in both $\mathcal{S}$ and $\mathcal{X}$. Denoting $\lambda \subseteq \mathcal{X}$ as the set of end states after re-partitioning, the agent can ground the operator by appending $\kappa$ to the precondition and $\lambda$ to the effect (if it differs from $\kappa$), where $\kappa$ and $\lambda$ are treated as problem-specific propositions. Finally, these problem-specific propositions must be linked with the grounded objects being acted upon. The agent therefore adds a precondition predicate conditioned on the identity of the grounded objects (see Appendix G for examples).

## 4 EXPERIMENTS

We first demonstrate our framework on the classic Blocks World domain (Section 4.1). While the high-level operators and predicates describing the domain are usually given, we show how such a representation can be learned autonomously from scratch. We then demonstrate that our method scales to significantly harder problems by applying it to a high-dimensional Minecraft task (Section 4.2). Finally, we investigate the transferability of the learned abstractions by transferring them to additional procedurally-generated Minecraft tasks (Section 4.3). Owing to space constraints, we defer the exact implementation and domain details to the appendix.

### 4.1 LEARNING A REPRESENTATION OF BLOCKS WORLD

The Blocks World domain consists of a number of blocks which can be stacked on top of one another by an agent (hand). The agent possesses options that allow it to pick up a block (`Pick`), put a block back on the table (`Put`), and stack one block on another (`Stack`). Blocks cannot be picked up if they are covered or if the hand is occupied, and can only be put down or stacked if already gripped. We consider the task consisting of three blocks `A`, `B` and `C`, where each block is described by whether there is nothing, another block, or a table directly above or below it. This representation allows us to determine whether a given block is on a table, on another block, or grasped in the hand, and similarly whether another block has been stacked upon it. The hand is characterised by a single boolean indicating whether it is holding a block. Thus a state is described by $\{\mathbf{f}_H, \mathbf{f}_A, \mathbf{f}_B, \mathbf{f}_C\}$, corresponding to the hand and blocks' features respectively. Note that the agent is initially unaware that the blocks are functionally identical and can be treated interchangeably.

**Generating a Propositional Model (Steps 1–2)**   Using the approach outlined in Section 3.1, the agent partitions the options using transition data collected from the environment. This results in a total of 15 partitions of the `Pick` option, 3 partitions of the `Put` option, and 12 partitions of the `Stack` option.[3] It then fits a classifier to each partition's initiation states, and a density estimator to its terminating states. Finally, the agent generates a propositional PDDL using these learned preconditions and effects. Figure 2 illustrates a learned propositional operator, while the full PDDL, learned entirely from data, is provided in Appendix B.

**Generating a Lifted Typed Model (Steps 3–4)**   Using the effects from the propositional representation, the agent determines that objects `A`, `B` and `C` all possess the same effect profiles for all options and so can be grouped into a single type, while the hand belongs to its own type. The agent can now lift its representation by replacing the learned propositions with predicates parameterised by

---

[3] See Appendix A for a description of each partitioned option.

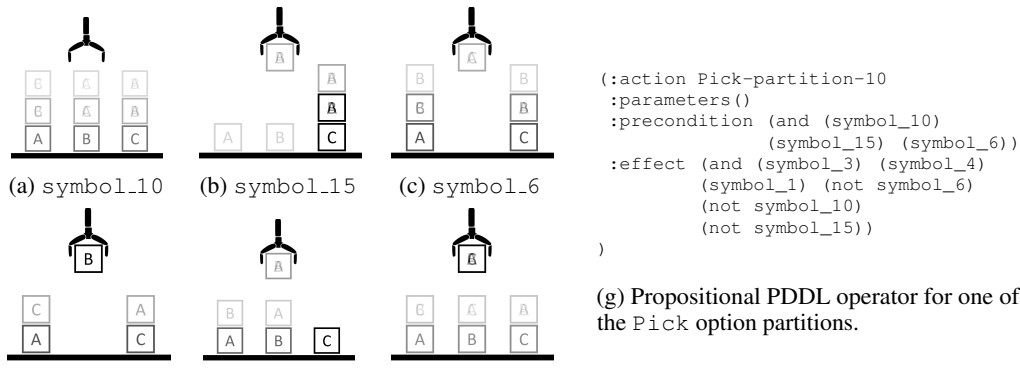

```
(:action Pick-partition-10
 :parameters()
 :precondition (and (symbol_10)
            (symbol_15) (symbol_6))
 :effect (and (symbol_3) (symbol_4)
        (symbol_1) (not symbol_6)
        (not symbol_10)
        (not symbol_15))
)
```

(g) Propositional PDDL operator for one of the `Pick` option partitions.

Figure 2: The learned propositional operator for a `Pick` action describing picking `B` off `C`. In order to execute the action, the hand must be empty (`symbol_10`), `C` must be on the table and covered by a block (`symbol_15`), and `B` must be on top of a block and uncovered (`symbol_6`). After execution, `B` is in the hand (`symbol_3`), `C` is on the table and clear (`symbol_4`), and the hand is full (`symbol_1`). We visualise the symbols by sampling from the propositional symbol, and randomly sampling the remaining independent state variables (since each symbol is a distribution over a subset of state variables). The transparency is due to the averaging over the independent state variables. Note that we must learn one operator for every pair of blocks.

the above types. For example, after generating the model, there are three propositions: `AOnTable`, `BOnTable`, and `COnTable`. Since these are distributions over objects determined to be the same type, the agent can replace them all with a single predicate `OnTable(X)`, which accepts block objects. As a result, the agent can reduce the number of operators from 30 to 6, resulting in a more compact representation with a smaller branching factor. Figure 3 illustrates how the propositional operator in Figure 2 has been lifted to describe picking any block `X` off any block `Y`.

The number of learned operators is a function of the number of blocks in the domain. Since the propositional approach treats each block as its own unique object, it must learn the dynamics and interaction of each new block it encounters. For $n$ blocks, this requires $O(n^2)$ operators. However, once the agent has learned the object types and constructed predicates based on these, it needs at most 6 operators to represent the dynamics for any number of blocks. See Appendix C for the full parameterised PDDL.

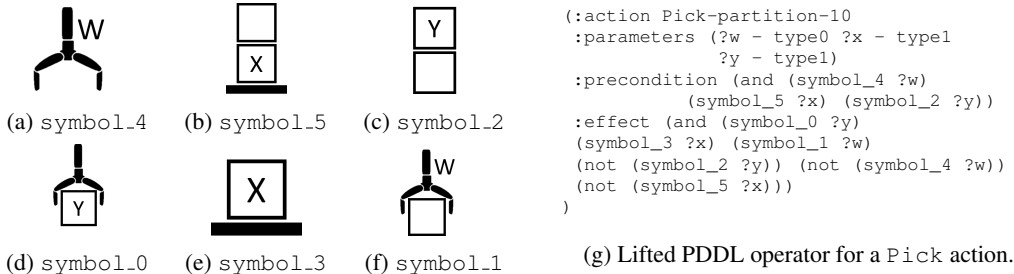

```
(:action Pick-partition-10
 :parameters (?w - type0 ?x - type1
            ?y - type1)
 :precondition (and (symbol_4 ?w)
        (symbol_5 ?x) (symbol_2 ?y))
 :effect (and (symbol_0 ?y)
 (symbol_3 ?x) (symbol_1 ?w)
 (not (symbol_2 ?y)) (not (symbol_4 ?w))
 (not (symbol_5 ?x)))
)
```

(g) Lifted PDDL operator for a `Pick` action.

Figure 3: The learned lifted operator for a `Pick` action describing picking a block off another. In order to pick up block `Y`, it must be on block `X` which itself is on the table, and the hand must be empty. As a result, the hand is not empty, `Y` is now in the hand, and `X` is on the table and clear. `type0` refers to the "hand" type, while `type1` refers to the "block" type.

## 4.2 Learning a Representation of a Minecraft Task

In the above example, objects were represented using pre-specified features and were sufficient to describe the environment dynamics. However, our approach is capable of scaling beyond this simple case and learning these features from pixels. We now demonstrate this in a complex Minecraft task

(Johnson et al., 2016) consisting of five rooms with various items positioned throughout. Rooms are connected with either regular doors which can be opened by direct interaction, or puzzle doors which require the agent to pull a lever to open. The world is described by the state of each of the objects (given directly by each object's appearance as a $600 \times 800$ RGB image), the agent's view, and current inventory. To simplify learning, we compress the state space by downscaling images and applying PCA (Pearson, 1901) to a greyscaled version, preserving the top 40 principal components.

The agent is given high-level skills, such as `ToggleDoor` and `WalkToItem`. Execution is stochastic—opening doors occasionally fails, and the navigation skills are noisy in their execution. To solve the task, an agent must first collect the pickaxe, use it to break the gold and redstone blocks and collect the resulting items. It must then navigate to the crafting table, where it uses the collected items to first craft gold ingots and subsequently a clock. Finally, it must navigate to the chest and open it to complete the task. This requires a long-horizon, hierarchical plan—the shortest plan that solves the task consists of 28 options consisting of *hundreds* of low-level continuous actions.

**Generating a Propositional Model (Steps 1–2)**    As previously, the agent begins by learning a model using the method outlined in Section 3.1 and in prior work (Konidaris et al., 2018; Ames et al., 2018). The agent partitions options using DBSCAN (Ester et al., 1996) to cluster option data based on terminating states. For each partitioned option, it then fits an SVM (Cortes & Vapnik, 1995) with Platt scaling (Platt, 1999) to estimate the preconditions, and a kernel density estimator (Rosenblatt, 1956) for effects, which are then used to construct the propositional PPDDL.

**Generating a Lifted, Typed Model (Steps 3–4)**    Using the effects from the propositional representation, the agent next groups objects into types based on their effect profiles. This is made easier because certain objects do not undergo effects under certain options. For example, the chest cannot be toggled, while a door can, and thus it is immediately clear that they are not of the same type. Having determined the types, the agent replaces all similar propositions (where similarity is measured using the KL-divergence) with a single predicate parameterised by an object of that type.

**Problem-Specific Instantiation (Step 5)**    The agent now has a representation whose operators can be transferred between tasks. However, unlike Blocks World, a complication arises because the object-centric state space is *not* Markov. For example, a state where all the doors are closed and the agent is in front of the first door is indistinguishable from a state where the agent is in front of the second door. As described in Section 3.3, the agent must ground the representations in the current task by incorporating additional problem-specific state variables to preserve the Markov property. These state variables are fixed across the family of MDPs; in this case, they are the agent's $xyz$-location.

For each partitioned option, the agent again uses DBSCAN to cluster end states $\mathcal{X}$ to form partitioned subgoal options in both $\mathcal{S}$ and $\mathcal{X}$. Each of these clusters in $\mathcal{X}$ is a problem-specific proposition, which can be added to the learned operators to ground the problem. In Figure 4, we illustrate a learned operator for opening a particular door, where the problem-specific symbol has been tied to the door being opened in this manner. Without modifying the operator's parameter, it would be possible to open *any* door at that location. The final plan discovered by the agent is illustrated by Figure 5.

## 4.3    Inter-task Transfer in Minecraft

We next investigate transferring operators between five procedurally-generated tasks, where each task differs in the location of the objects and doors; the agent cannot thus simply use a plan found in one task to solve another. For a given task, the agent transfers all operators learned from previous tasks, and continues to collect samples using uniform random exploration until it produces a model which predicts that the optimal plan can be executed. Figure 6a shows the operators transferred between tasks, while Figure 6b shows the number of interactions required to learn a model in a new task.

The minimum number of samples required to learn a model for a new task is bounded by the exploration strategy, since we must discover all problem-specific symbols to complete the model. Figure 6b shows that the number of samples required to learn a model decreases over time towards this lower bound. Inter-task transfer could be further improved by leveraging the agent's existing knowledge to perform non-uniform exploration, but we leave this to future work.

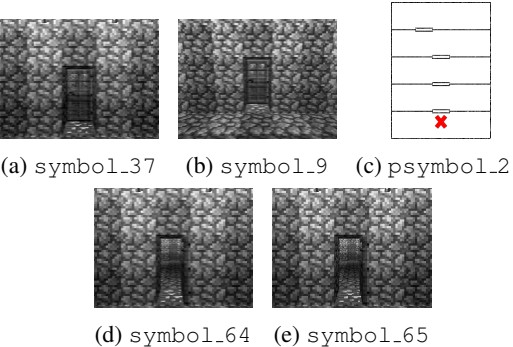

(a) `symbol_37` (b) `symbol_9` (c) `psymbol_24`

(d) `symbol_64` (e) `symbol_65`

```
(:action Toggle-Door-partition-1a
:parameters (?w - type0 ?x - type1)
:precondition (and (notfailed)
        (symbol_37 ?w) (symbol_9 ?x)
        (= (id ?x) 1) (psymbol_24))
:effect (and (symbol_64 ?x)
    (symbol_65 ?w) (not (symbol_9 ?x))
    (not (symbol_37 ?w)))
)
```

(f) A learned typed PDDL operator for one partition of the `Toggle-Door` option. The predicates underlined in red must be re-learned for each new task, while the rest of the operator can be safely transferred.

Figure 4: Our approach learns that, in order to open a particular door, the agent must be standing in front of a closed door (`symbol_37`) at a particular location (`psymbol_24`), and the door must be closed (`symbol_9`). The effect of the skill is that the agent finds itself in front of an open door (`symbol_64`) and the door is open (`symbol_65`). `type0` and `type1` refer to the "agent" and "door" classes, while `id` is a fluent specifying the identity of the grounded door object, and is linked to the problem-specific symbol underlined in red.

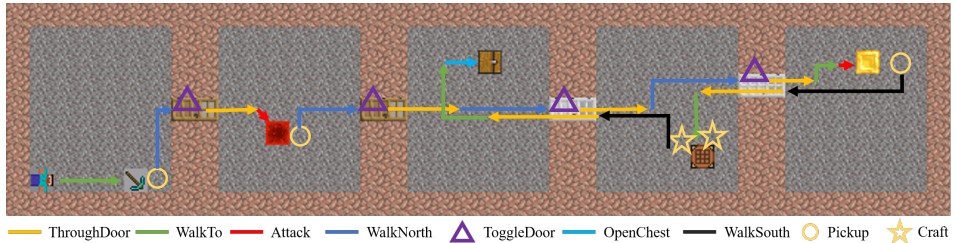

Figure 5: Path traced by the agent executing different options while solving the first task.

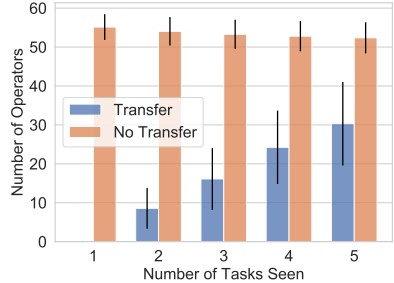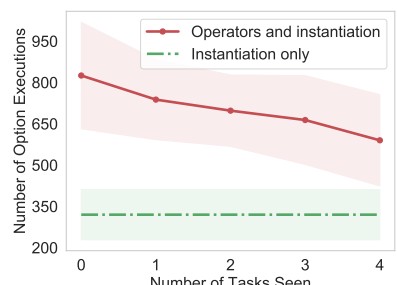

(a) Orange bars represent the number of operators that must be learned to produce a sufficiently accurate model to solve the task. Blue bars represent the number of operators transferred between tasks. As the number of tasks increases, the number of new operators that must be learned decreases.

(b) Number of samples required to learn sufficiently accurate models as a function of the number of tasks encountered. The red line represents the number of samples required to learn all the operators and the instantiation, while the green line accounts for the instantiation phase only.

Figure 6: Results of learning and transferring high-level abstractions between tasks. We report the mean and standard deviation averaged over 80 runs with random task orderings.

## 5 RELATED WORK AND CONCLUSION

There has been work autonomously learning parameterised, transferable representations of skills. Ugur & Piater (2015) learn object-centric PDDL representations for robotic object manipulation tasks. They are able to learn object types which are similar to ours directly from data, but the object features are specified prior to learning, and discrete relations between object properties such as

width and height are given. Similarly, certain predicates are manually inserted to generate a sound representation. Asai (2019) learns object-centric abstractions directly from pixels, but it is unclear how to extend the approach to the stochastic setting. Furthermore, the representations lack soundness guarantees, and cannot be transformed into a language that can be used by existing task-level planners. Finney et al. (2002), Pasula et al. (2004) and Zettlemoyer et al. (2005) are able to learn operators that transfer across tasks, but the high-level symbols that constitute the state space are given. James et al. (2020) learn a PPDDL representation for planning using agent-relative and problem-specific data, but the operators are not lifted and there is no notion of objects or types, which limit generalisability.

Alternatively, object-oriented MDPs (Diuk et al., 2008) specify the state space as a set of objects belonging to classes with associated attributes. The agent must learn the transition dynamics, which are usually restricted to a small number of effects. We show how to learn an object-centric representation along with the object types, as well as the abstract high-level dynamics model.

By contrast, we have shown how to learn the vocabulary *and* the type system *and* the model from raw sensory input. Our representations are useful for planning, generalise across objects and can be transferred to new tasks. Although we have injected structure by assuming the existence of objects, this reflects the nature of many environments: fields such as computer vision assume that the world consists of objects, while there is evidence to suggest that infants do the same (Spelke, 1990). This assumption allows us to convert complex, high-dimensional environments to abstract representations that serve as input to task-level planners. Our approach provides an avenue for solving sparse-reward, long-term planning problems—such as the MineRL competition (Guss et al., 2019)—currently beyond the reach of model-free approaches.

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

## A    ENUMERATING SUBGOAL OPTIONS FOR THE BLOCKS WORLD DOMAIN

Given the description of the Blocks World domain in the main text, we must partition the given options (`Pick`, `Put` and `Stack`) so that they adhere to the subgoal condition. When there are three blocks in the environment, we see that there are 30 partitioned options, which are described in the table below

Table 1: Descriptions of the different option partitions. The description of start and end states includes only the relevant information.

| Option | # partitions | Description of start states | Description of end states |
|---|---|---|---|
| `PickOffTable(X)` | 3 | X is on the table, X is clear, and the hand is empty. | X is grasped in the hand. |
| `PickOffSingleBlock(X, Y)` | 6 | X is on block Y which is on the table, X is clear, and the hand is empty. | X is grasped in the hand and Y is clear and on the table. |
| `PickOffDoubleBlock(X, Y)` | 6 | X is on block Y which is on another block, X is clear, and the hand is empty. | X is grasped in the hand and Y is clear and on another block. |
| `StackOnSingleBlock(X, Y)` | 6 | X is in the hand, and Y is clear and on the table. | X is on block Y which is on the table, and the hand is empty. |
| `StackOnDoubleBlock(X, Y)` | 6 | X is in the hand, and Y is clear and on another block. | X is on block Y which is on another block, and the hand is empty. |
| `Put(X)` | 3 | X is grasped in the hand. | X is on the table and the hand is empty. |

## B    PROPOSITIONAL PDDL DESCRIPTION FOR THE BLOCKS WORLD TASK

Below is the automatically generated propositional PDDL description of the Blocks World domain with 3 blocks. In practice, the agent generates this description with arbitrary names for the propositions, but for readability purposes we have manually renamed them to match their semantics.

```
(define (domain BlocksWorld)
        (:requirements :strips)
        (:predicates
                (notfailed)
                (AInHand)
                (HandFull)
                (COnBlock)
                (BInHand)
                (COnTable)
                (AOnTable)
                (BOnBlock)
                (AOnBlock)
                (BOnTable)
                (CInHand)
                (HandEmpty)
                (BOnTable_BCovered)
                (COnBlock_CCovered)
                (AOnBlock_ACovered)
                (BOnBlock_BCovered)
                (COnTable_CCovered)
                (AOnTable_ACovered)
        )
```

```
(:action Pick_0
 :parameters()
 :precondition (and (HandEmpty) (AOnTable) (notfailed))
 :effect (and (AInHand) (HandFull) (not AOnTable) (not HandEmpty) (not AOnTable)
         (not HandEmpty))
)

(:action Pick_1
 :parameters()
 :precondition (and (HandEmpty) (AOnBlock) (COnBlock_CCovered) (notfailed))
 :effect (and (COnBlock) (AInHand) (HandFull) (not AOnBlock) (not HandEmpty)
         (not COnBlock_CCovered) (not AOnBlock) (not HandEmpty)
         (not COnBlock_CCovered))
)

(:action Pick_2
 :parameters()
 :precondition (and (HandEmpty) (COnTable_CCovered) (BOnBlock) (notfailed))
 :effect (and (BInHand) (COnTable) (HandFull) (not BOnBlock) (not HandEmpty)
         (not COnTable_CCovered) (not BOnBlock) (not HandEmpty)
         (not COnTable_CCovered))
)

(:action Pick_3
 :parameters()
 :precondition (and (HandEmpty) (AOnTable_ACovered) (BOnBlock) (notfailed))
 :effect (and (BInHand) (AOnTable) (HandFull) (not BOnBlock) (not HandEmpty)
         (not AOnTable_ACovered) (not BOnBlock) (not HandEmpty)
         (not AOnTable_ACovered))
)

(:action Pick_4
 :parameters()
 :precondition (and (HandEmpty) (AOnBlock) (BOnBlock_BCovered) (notfailed))
 :effect (and (BOnBlock) (AInHand) (HandFull) (not AOnBlock) (not HandEmpty)
         (not BOnBlock_BCovered) (not AOnBlock) (not HandEmpty)
         (not BOnBlock_BCovered))
)

(:action Pick_5
 :parameters()
 :precondition (and (HandEmpty) (AOnBlock_ACovered) (BOnBlock) (notfailed))
 :effect (and (BInHand) (AOnBlock) (HandFull) (not BOnBlock) (not HandEmpty)
         (not AOnBlock_ACovered) (not BOnBlock) (not HandEmpty)
         (not AOnBlock_ACovered))
)

(:action Pick_6
 :parameters()
 :precondition (and (HandEmpty) (AOnBlock) (BOnTable_BCovered) (notfailed))
 :effect (and (BOnTable) (AInHand) (HandFull) (not AOnBlock) (not HandEmpty)
         (not BOnTable_BCovered) (not AOnBlock) (not HandEmpty)
         (not BOnTable_BCovered))
)

(:action Pick_7
 :parameters()
 :precondition (and (HandEmpty) (BOnTable) (notfailed))
 :effect (and (BInHand) (HandFull) (not BOnTable) (not HandEmpty) (not BOnTable)
         (not HandEmpty))
)

(:action Pick_8
 :parameters()
 :precondition (and (HandEmpty) (COnTable) (notfailed))
 :effect (and (CInHand) (HandFull) (not COnTable) (not HandEmpty) (not COnTable)
         (not HandEmpty))
)

(:action Pick_9
 :parameters()
 :precondition (and (HandEmpty) (AOnTable_ACovered) (COnBlock) (notfailed))
 :effect (and (CInHand) (AOnTable) (HandFull) (not COnBlock) (not HandEmpty)
         (not AOnTable_ACovered) (not COnBlock) (not HandEmpty)
         (not AOnTable_ACovered))
)

(:action Pick_10
 :parameters()
 :precondition (and (HandEmpty) (AOnBlock) (COnTable_CCovered) (notfailed))
```

```
     :effect (and (COnTable) (AInHand) (HandFull) (not AOnBlock) (not HandEmpty)
            (not COnTable_CCovered) (not AOnBlock) (not HandEmpty)
            (not COnTable_CCovered))
)

(:action Pick_11
 :parameters()
 :precondition (and (HandEmpty) (AOnBlock_ACovered) (COnBlock) (notfailed))
 :effect (and (CInHand) (AOnBlock) (HandFull) (not COnBlock) (not HandEmpty)
            (not AOnBlock_ACovered) (not COnBlock) (not HandEmpty)
            (not AOnBlock_ACovered))
)

(:action Pick_12
 :parameters()
 :precondition (and (HandEmpty) (COnBlock) (BOnTable_BCovered) (notfailed))
 :effect (and (BOnTable) (CInHand) (HandFull) (not COnBlock) (not HandEmpty)
            (not BOnTable_BCovered) (not COnBlock) (not HandEmpty)
            (not BOnTable_BCovered))
)

(:action Pick_13
 :parameters()
 :precondition (and (HandEmpty) (COnBlock_CCovered) (BOnBlock) (notfailed))
 :effect (and (BInHand) (COnBlock) (HandFull) (not BOnBlock) (not HandEmpty)
            (not COnBlock_CCovered) (not BOnBlock) (not HandEmpty)
            (not COnBlock_CCovered))
)

(:action Pick_14
 :parameters()
 :precondition (and (HandEmpty) (COnBlock) (BOnBlock_BCovered) (notfailed))
 :effect (and (BOnBlock) (CInHand) (HandFull) (not COnBlock) (not HandEmpty)
            (not BOnBlock_BCovered) (not COnBlock) (not HandEmpty)
            (not BOnBlock_BCovered))
)

(:action Put_15
 :parameters()
 :precondition (and (HandFull) (AInHand) (notfailed))
 :effect (and (AOnTable) (HandEmpty) (not AInHand) (not HandFull) (not AInHand)
            (not HandFull))
)

(:action Put_16
 :parameters()
 :precondition (and (HandFull) (BInHand) (notfailed))
 :effect (and (BOnTable) (HandEmpty) (not HandFull) (not BInHand) (not HandFull)
            (not BInHand))
)

(:action Put_17
 :parameters()
 :precondition (and (HandFull) (CInHand) (notfailed))
 :effect (and (COnTable) (HandEmpty) (not HandFull) (not CInHand) (not HandFull)
            (not CInHand))
)

(:action Stack_18
 :parameters()
 :precondition (and (HandFull) (CInHand) (BOnTable) (notfailed))
 :effect (and (BOnTable_BCovered) (COnBlock) (HandEmpty) (not HandFull)
            (not BOnTable) (not CInHand) (not HandFull) (not BOnTable)
            (not CInHand))
)

(:action Stack_19
 :parameters()
 :precondition (and (HandFull) (COnBlock) (BInHand) (notfailed))
 :effect (and (BOnBlock) (COnBlock_CCovered) (HandEmpty) (not HandFull)
            (not COnBlock) (not BInHand) (not HandFull) (not COnBlock)
            (not BInHand))
)

(:action Stack_20
 :parameters()
 :precondition (and (HandFull) (AOnBlock) (BInHand) (notfailed))
 :effect (and (BOnBlock) (AOnBlock_ACovered) (HandEmpty) (not HandFull)
            (not BInHand) (not AOnBlock) (not HandFull) (not BInHand)
            (not AOnBlock))
)
```

```
(:action Stack_21
 :parameters()
 :precondition (and (HandFull) (AInHand) (BOnTable) (notfailed))
 :effect (and (BOnTable_BCovered) (AOnBlock) (HandEmpty) (not AInHand)
         (not HandFull) (not BOnTable) (not AInHand) (not HandFull)
         (not BOnTable))
)

(:action Stack_22
 :parameters()
 :precondition (and (HandFull) (CInHand) (BOnBlock) (notfailed))
 :effect (and (BOnBlock_BCovered) (COnBlock) (HandEmpty) (not HandFull)
         (not BOnBlock) (not CInHand) (not HandFull) (not BOnBlock)
         (not CInHand))
)

(:action Stack_23
 :parameters()
 :precondition (and (HandFull) (COnTable) (BInHand) (notfailed))
 :effect (and (BOnBlock) (COnTable_CCovered) (HandEmpty) (not HandFull)
         (not BInHand) (not COnTable) (not HandFull) (not BInHand)
         (not COnTable))
)

(:action Stack_24
 :parameters()
 :precondition (and (HandFull) (AInHand) (COnBlock) (notfailed))
 :effect (and (COnBlock_CCovered) (AOnBlock) (HandEmpty) (not AInHand)
         (not HandFull) (not COnBlock) (not AInHand) (not HandFull)
         (not COnBlock))
)

(:action Stack_25
 :parameters()
 :precondition (and (HandFull) (AOnTable) (CInHand) (notfailed))
 :effect (and (COnBlock) (AOnTable_ACovered) (HandEmpty) (not HandFull)
         (not AOnTable) (not CInHand) (not HandFull) (not AOnTable)
         (not CInHand))
)

(:action Stack_26
 :parameters()
 :precondition (and (HandFull) (AInHand) (COnTable) (notfailed))
 :effect (and (COnTable_CCovered) (AOnBlock) (HandEmpty) (not AInHand)
         (not HandFull) (not COnTable) (not AInHand) (not HandFull)
         (not COnTable))
)

(:action Stack_27
 :parameters()
 :precondition (and (HandFull) (AOnBlock) (CInHand) (notfailed))
 :effect (and (COnBlock) (AOnBlock_ACovered) (HandEmpty) (not HandFull)
         (not AOnBlock) (not CInHand) (not HandFull) (not AOnBlock)
         (not CInHand))
)

(:action Stack_28
 :parameters()
 :precondition (and (HandFull) (AOnTable) (BInHand) (notfailed))
 :effect (and (BOnBlock) (AOnTable_ACovered) (HandEmpty) (not HandFull)
         (not BInHand) (not AOnTable) (not HandFull) (not BInHand)
         (not AOnTable))
)

(:action Stack_29
 :parameters()
 :precondition (and (HandFull) (AInHand) (BOnBlock) (notfailed))
 :effect (and (BOnBlock_BCovered) (AOnBlock) (HandEmpty) (not AInHand)
         (not HandFull) (not BOnBlock) (not AInHand) (not HandFull)
         (not BOnBlock))
)
)
```

## C    LIFTED PDDL DESCRIPTION FOR THE BLOCKS WORLD TASK

In contrast, the lifted representation learned below is far more compact. Here, operators are parameterised by objects, which allows for better generalisation across instances with varying numbers of blocks. Figure 7 shows how the number of partitioned options (and hence number of preconditions and effects) scales with the number of objects in the domain.

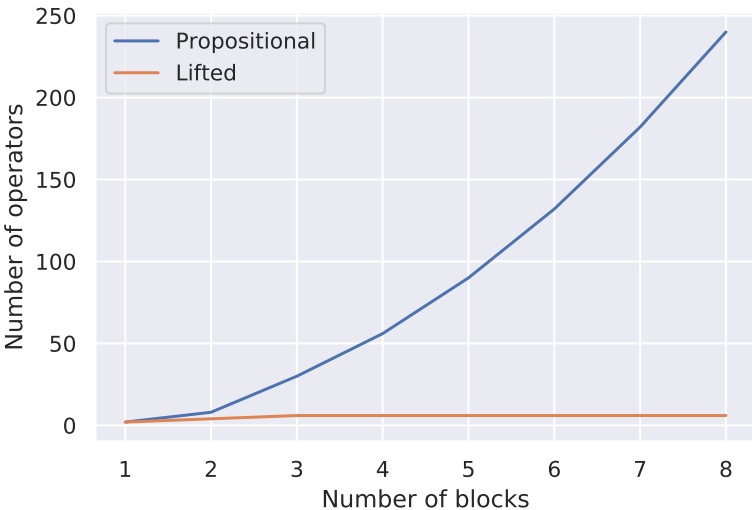

Figure 7: The number of learned action operators as a function of the number of blocks in the domain. Since the propositional approach treats each block as its own, unique object, it must learn the dynamics and interaction of each new block it encounters. For $n$ blocks, this requires $O(n^2)$ operators. However, if we learn the object types and construct predicates based on these, then we need at most 6 operators to represent the dynamics for any number of blocks.

Below we provide the learned representation for the domain. Again, we manually rename the predicates and types to help with readability.

```
(define (domain BlocksWorld)
        (:requirements :strips :typing)
        (:types hand block)
        (:predicates
                (BlockInHand ?w - block)
                (HandFull ?w - hand)
                (BlockOnBlock ?w - block)
                (BlockOnTable ?w - block)
                (HandEmpty ?w - hand)
                (BlockOnTable_BlockCovered ?w - block)
                (BlockOnBlock_BlockCovered ?w - block)
                (notfailed)
        )
        (:action Pick-partition-0
         :parameters (?w - hand ?x - block)
         :precondition (and (notfailed) (HandEmpty ?w) (BlockOnTable ?x))
         :effect (and (BlockInHand ?x) (HandFull ?w) (not (BlockOnTable ?x))
                (not (HandEmpty ?w)))
        )

        (:action Pick-partition-1
         :parameters (?w - hand ?x - block ?y - block)
         :precondition (and (notfailed) (HandEmpty ?w) (BlockOnBlock ?x)
                        (BlockOnBlock_BlockCovered ?y))
         :effect (and (BlockOnBlock ?y) (BlockInHand ?x) (HandFull ?w)
                (not (BlockOnBlock ?x)) (not (HandEmpty ?w))
                (not (BlockOnBlock_BlockCovered ?y)))
        )

        (:action Pick-partition-10
         :parameters (?w - hand ?x - block ?y - block)
```

```
            :precondition (and (notfailed) (HandEmpty ?w) (BlockOnTable_BlockCovered ?x)
                             (BlockOnBlock ?y))
            :effect (and (BlockInHand ?y) (BlockOnTable ?x) (HandFull ?w)
                         (not (BlockOnBlock ?y)) (not (HandEmpty ?w))
                         (not (BlockOnTable_BlockCovered ?x)))
        )

        (:action Put-partition-0
         :parameters (?w - hand ?x - block)
         :precondition (and (notfailed) (HandFull ?w) (BlockInHand ?x))
         :effect (and (BlockOnTable ?x) (HandEmpty ?w) (not (BlockInHand ?x))
                      (not (HandFull ?w)))
        )

        (:action Stack-partition-0
         :parameters (?w - hand ?x - block ?y - block)
         :precondition (and (notfailed) (HandFull ?w) (BlockInHand ?x) (BlockOnTable ?y))
         :effect (and (BlockOnTable_BlockCovered ?y) (BlockOnBlock ?x) (HandEmpty ?w)
                      (not (HandFull ?w)) (not (BlockOnTable ?y))
                      (not (BlockInHand ?x)))
        )

        (:action Stack-partition-1
         :parameters (?w - hand ?x - block ?y - block)
         :precondition (and (notfailed) (HandFull ?w) (BlockOnBlock ?x) (BlockInHand ?y))
         :effect (and (BlockOnBlock ?y) (BlockOnBlock_BlockCovered ?x) (HandEmpty ?w)
                      (not (HandFull ?w)) (not (BlockOnBlock ?x))
                      (not (BlockInHand ?y)))
        )

)
```

A task might then be specified as follows:

```
(define (problem stack)
    (:domain BlocksWorld)

    (:objects hand - Hand
            A B C - Block
    )
    (:init (BlockOnTable A)
           (BlockOnTable B)
           (BlockOnTable C)
           (HandEmpty hand)
           (notfailed)
     )
    (:goal (and (BlockOnBlock A)
                (BlockOnBlock_BlockCovered C)
                (BlockOnTable_BlockCovered B)))
)
```

## D  MINECRAFT TASK DETAILS

Our Minecraft tasks are procedurally generated, consisting of five rooms with various items positioned throughout. Rooms are connected with either regular doors which can be opened by direct interaction, or puzzle doors which require the agent to pull a lever to open. The world is described by the state of each of the objects (given directly by each object's appearance as a $600 \times 800$ RGB image), the agent's view, and current inventory. Figure 8 illustrates the state of each object in the world at the beginning of one of the tasks.

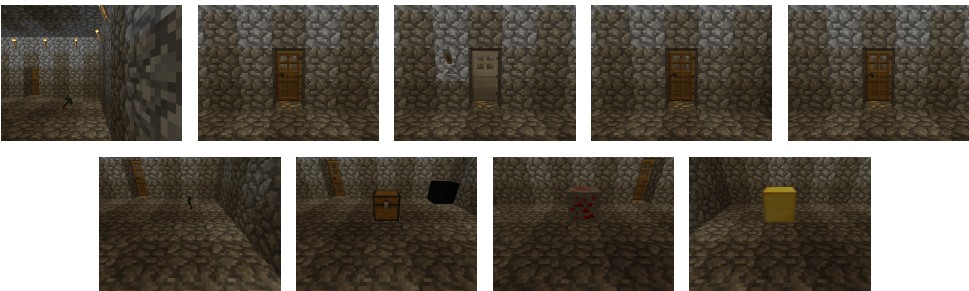

Figure 8: The state of each object in the world at the start of the task. From left to right, the images represent the agent's point of view, the four doors, the pickaxe, the chest, and the redstone and gold blocks. The inventory is not shown here.

The agent is provided with the following high-level skills:

(i) `WalkToItem`—the agent will approach an item if it is in the same room.

(ii) `AttackBlock`—the agent will break a block, provided it is near the block and holding the pickaxe.

(iii) `PickupItem`—the agent will collect the item if it is standing in front of it.

(iv) `WalkToNorthDoor`—the agent will approach the northern door in the current room.

(v) `WalkToSouthDoor`—the agent will approach the southern door in the current room.

(vi) `WalkThroughDoor`—the agent will walk through a door to the next room, provided the door is open.

(vii) `CraftItem`—the agent will create a new item from ingredients in its inventory, provided it is near the crafting table.

(viii) `OpenChest`—the agent will open the chest, provided it is standing in front of it and possesses the clock.

(ix) `ToggleDoor`—the agent will open or close the door directly in front of it.

Execution is stochastic—opening doors occasionally fails, and the navigation skills are noisy in their execution.

## E  LEARNING A PORTABLE REPRESENTATION FOR MINECRAFT

In this section, we describe the exact details for learning a representation of a Minecraft task. Pseudocode for the approach (independent of the domain) is provided in Section F.

In order to learn a high-level representation, we first apply a series preprocessing steps to reduce the dimensionality of the state space. We downscale images to $160 \times 120$ and then convert the resulting images to greyscale. We apply principal component analysis (Pearson, 1901) to a batch of images collected from the different tasks and keep the top 40 principal components. This allows us to represent each object (except the inventory, which is a one-hot encoded vector of length 5) as a vector of length 40.

**Partitioning** We collect data from a task by executing options uniformly at random. We record state transition data as well as, for each state, which options could be executed. We then partition options using the DBSCAN clustering algorithm (Ester et al., 1996) to cluster the terminating states of each option into separate effects. This approximately preserves the subgoal property, as described in Section 2 and previous work (Andersen & Konidaris, 2017; Konidaris et al., 2018; Ames et al., 2018). For each pair of partitioned options, we check whether their is significant overlap in their initiating states (again using DBSCAN). If the initiating states overlap significantly, the partitions are merged to account for probabilistic effects.

**Preconditions** Next, the agent learns a precondition classifier for each of these approximately partitioned options using an SVM (Cortes & Vapnik, 1995) with Platt scaling (Platt, 1999). We use states initially collected as negative examples, and data from the actual transitions as positive examples. We employ a simple feature selection procedure to determine which objects are relevant to the option's precondition. We first compute the accuracy of the SVM applied to the object the option operates on, performing a grid search to find the best hyperparameters for the SVM using 3-fold cross validation. Then, for every other object in the environment, we compute the SVM's accuracy when that object's features are added to the SVM. Any object that increases the SVM accuracy is kept. Pseudocode for this procedure is outline in Figure 9.

Having determined the relevant objects, we fit a probabilistic SVM to the relevant objects' data. Note that we learn a single SVM for a given precondition. Thus if the precondition includes two objects, then the SVM will learn a classifier over both objects' features jointly.

```
 1: procedure FEATURESELECTION
 2:     Given: affected objects Mask, positive start states p, negative start states n, set of objects M
 3:     ▷ Fit a classifier over only objects in the mask
 4:     classifier ← FITCLASSIFIER(start, negative, mask)
 5:     initScore ← classifier.score
 6:     Keep ← ∅
 7:     for each object ∈ M \ Mask do
 8:         classifier ← FITCLASSIFIER(start, negative, Mask ∪ {object})
 9:         newScore ← classifier.score
10:         if newScore > initScore then
11:             ▷ Keep the object if it improves the score
12:             Keep ← Keep ∪ {object}
13:         end if
14:     end for
15:     return Mask ∪ Keep
16: end procedure
```

Figure 9: Pseudocode for a simple feature selection procedure.

**Effects** A kernel density estimator (KDE) (Rosenblatt, 1956) with Gaussian kernel is used to estimate the effect of each partitioned option. We learn distributions over only the objects affected by the option, learning one KDE for each object. We use a grid search with 3-fold cross validation to find the best bandwidth hyperparameter for each estimator. We fit a single KDE to each object separately, since the state space has already been factored into these objects. Each of these KDEs is an abstract symbol in our propositional PDDL representation.

**Propositional PDDL** For each partitioned option, we now have a classifier and set of effect distributions (propositions). However, to generate the PDDL, the precondition must be specified in terms of these propositions. We use the same approach as Konidaris et al. (2018) to generate the PDDL: for all combinations of valid effect distributions, we test whether data sampled from their conjunction is evaluated positively by our classifiers. If they are, then that combination of distributions serves as the precondition of the high-level operator. This procedure is described in Figure 10.

```
 1: procedure BUILDPPDDLOPERATOR
 2:     Given: precondition classifier classifier, current effect effect, all effects Effects
 3:     Operators ← ∅
 4:     Symbols ← ∅
 5:     for each candidate ∈ ℘(Effects) do              ▷ For all possible effect combinations
 6:         samples ← SAMPLE(candidate)                      ▷ Sample from the distributions
 7:         prob ← PREDICT(classifier, sample)               ▷ Query the classifier with the data
 8:         if prob > 0 then
 9:             if prob = 1 then
10:                 ▷ Construct the new operator with the existing effects
11:                 operator ← {candidate, effect}
12:             else
13:                 ▷ Add a probabilistic failure case
```

$$14: \quad newEffect \leftarrow \begin{cases} \texttt{fail}, \text{with probability } (1 - prob) \\ effect, \text{with probability } prob \end{cases}$$

```
15:                 operator ← {candidate, newEffect}
16:             end if
17:             Operators ← Operators ∪ {operator}
18:             Symbols ← Symbols ∪ {candidate} ∪ {effect}
19:         end if
20:     end for
21:     return Operators, Symbols
22: end procedure
```

Figure 10: Pseudocode for constructing propositional PPDDL operators.

**Type Inference** To determine the type of each object, we first assume that they all belong to their own type. For each object, we compute its effect profile by extracting the effect propositions that occur under each option. Figure 11 illustrates this process.

For each pair of objects, we then determine whether the effect profiles are similar. This task is made easier because certain objects do not undergo effects with certain options. For example, the gold block cannot be toggled, while a door can. Thus it is easy to see that they are not of the same type. To determine whether two distributions are similar, we simply check whether the KL-divergence is less than a certain threshold. Having determined the types, we can simply replace all similar propositions with a predicate parameterised by an object of that type, as described by Figure 12.

```
 1: procedure COMPUTEEFFECTS
 2:     Given: object i, option o, PPDDL operators Operators
 3:     ▷ Get only the operators that model option o
 4:     Operators ← {operator | ∀operator ∈ Operators, REFERSTO(operator, o)}
 5:     Effects ← ∅
 6:     for each {·, effect} ∈ Operators do
 7:         ▷ Extract the effect propositions that refer to distributions over object i
 8:         OperatorEffect ← {prop | ∀prop ∈ effect, REFERSTO(prop, i)}
 9:         Effects ← Effects ∪ {OperatorEffect}
10:     end for
11:     return Effects
12: end procedure
```

Figure 11: Pseudocode for computing the effect distributions under an option for a given object.

```
 1: procedure MERGE
 2:     Given: objects M, type T, PPDDL operators Operators, propositions Propositions
 3:     ▷ Find the first object matching the type
 4:     archetype ← ∅
 5:     for each object ∈ M do
 6:         if ISTYPE(object, T) then
 7:             archetype ← object
 8:             break
 9:         end if
10:     end for
11:     ▷ Remove propositions with objects of type T that are not the archetype
12:     Removed ← {prop | ∀prop ∈ Propositions, ISTYPE(prop, T),
                       ¬REFERSTO(prop, archetype)}
13:     ▷ Keep operators that do not contain the removed propositions
14:     Operators ← {op | ∀op ∈ Operators, Removed ∩ op = ∅}
15:     return Operators, Propositions \ Removed
16: end procedure
```

Figure 12: Pseudocode for lifting propositions to typed predicates.

**Problem-Specific Instantiation**    Finally, we again use DBSCAN to partition our subgoal options, but this time using problem-specific state variables. Each of these clusters is then added to our representation as a problem-specific proposition. To ground the operators, we add the start and end clusters (problem-specific propositions) to the precondition and effects of the PPDDL operator. We also record the grounded object that appears in the parameter list of each operator, and add a precondition predicate (fluent) to ensure that only *those* particular objects can be modified. Without this final step, the agent would, for example, believe it can open *any* door while standing in front of a door at a particular location. We have thus linked the particular door to a particular location in the domain.

## F    PSEUDOCODE

Below we present pseudocode describing our approach to building a typed, object-centric PPDDL representation for an arbitrary domain. Some subroutines used in the pseudocode below are outlined in the previous section.

```
 1: procedure LEARNREPRESENTATION
 2:     Given: T state-option transitions D = {(s_i, x_i, o_i, s'_i, x'_i) | 0 ≤ i ≤ T}, set of objects M
 3:     ▷ Partition options into subgoal options
 4:     SubgoalOptions ← ∅
 5:     for each o ∈ O do
 6:         I ← {s | (s, ·, o, ·, ·) ∈ D}                          ▷ Set of initial states for option o
 7:         β ← {s' | (·, ·, o, s', ·) ∈ D}                        ▷ Set of terminating states for option o
 8:         for all K ⊆ I such that Pr(s' | s_i, o) = Pr(s' | s_j, o)∀s_i, s_j ∈ I, s' ∈ β do
 9:             P ← {o, K, {s' | ∀s ∈ K, (s, ·, o, s', ·) ∈ D}}   ▷ Start and end states for a partition
10:             SubgoalOptions ← SubgoalOptions ∪ {P}
11:         end for
12:     end for
13:     ▷ Estimate preconditions and effects
14:     Preconditions, Effects ← ∅
15:     for each {·, start, end} ∈ SubgoalOptions do
16:         mask ← COMPUTEMASK(start, end)                         ▷ List the objects that change state
17:         negative ← S \ start
18:         features ← FEATURESELECTION(mask, start, negative)
19:         classifier ← FITCLASSIFIER(start, negative, features)
20:         Preconditions ← Preconditions ∪ {classifier}
21:         estimator ← FITESTIMATOR(mask, end)                    ▷ Fit over only objects that change
22:         Effects ← Effects ∪ {estimator}
```

```
23:     end for
24:        ▷ Build propositional PPDDL
25:        Operators, Propositions ← ∅
26:     for each precondition, effect ∈ (Preconditions × Effects) do
27:           op, symbols ← BUILDPPDDLOPERATOR(precondition, effect, Effects)
28:           Operators ← Operators ∪ {op}
29:           Propositions ← Propositions ∪ symbols
30:     end for
31:        ▷ Infer object types
32:        EffProfile ← ∅
33:     for each object m do
34:        for each o ∈ O do
35:              EffProfile(m, o) ← COMPUTEEFFECTS(m, o, Operators)
36:        end for
37:     end for
38:     Types ← {K | EffProfile(mᵢ, o) ≈ EffProfile(mⱼ, o)∀o ∈ O, mᵢ, mⱼ ∈ K, K ⊆ M}
39:        ▷ Generate typed PPDDL
40:        TypedOperators, Predicates ← ∅
41:     for each type ∈ Types do
42:           ▷ Replace propositions and operators over objects of same type with lifted versions
43:           ops, predicates ← MERGE(M, type, Operators, Propositions)
44:           TypedOperators ← TypedOperators ∪ ops
45:           Predicates ← Predicates ∪ predicates
46:     end for
47:        ▷ Instantiate typed PPDDL in new task
48:     for each {o, start, end} ∈ SubgoalOptions do
49:           I_X ← {x | ∀s ∈ start, s' ∈ end, (s, x, o, s', ·) ∈ D}
50:           β_X ← {x' | ∀s ∈ start, s' ∈ end, x ∈ I_X, (s, x, o, s', x') ∈ D}
51:        for all κ ⊆ I_X such that Pr(x' | xᵢ, o) = Pr(x' | xⱼ, o)∀xᵢ, xⱼ ∈ I_X, x' ∈ β_X do
52:              λ ← {x' | ∀s ∈ start, s' ∈ end, x ∈ κ, (s, x, o, s', x') ∈ D}
53:              Predicates ← Predicates ∪ {κ} ∪ {λ}          ▷ Add problem-specific symbols
54:              mask ← COMPUTEMASK(start, end)               ▷ Computes the affected objects
55:              ▷ Link problem-specific symbols in precondition and effect to the affected objects
56:              TypedOperators ← GROUND(TypedOperators, κ, λ, mask)
57:        end for
58:     end for
59:     return TypedOperators, Predicates
60: end procedure
```

## G    VISUALISING OPERATORS FOR MINECRAFT

Here we illustrate some learned operators for the Minecraft tasks. To see all predicates and operators, please see the following URL: `https://sites.google.com/view/mine-pddl`.

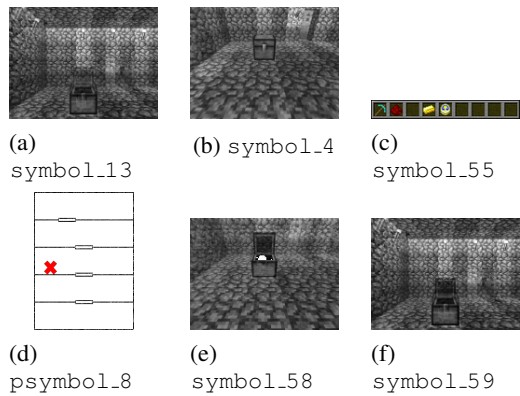

(a)
symbol_13

(b) symbol_4

(c)
symbol_55

(d)
psymbol_8

(e)
symbol_58

(f)
symbol_59

```
(:action Open-Chest-partition-0
:parameters (?w - type0 ?x - type6
              ?y - type9)
:precondition (and (notfailed)
          (symbol_13 ?w) (symbol_4 ?x)
          (symbol_55 ?y) (psymbol_8))
:effect (and (symbol_58 ?x) (symbol_59 ?w)
      (not (symbol_4 ?x))
      (not (symbol_13 ?w)))
)
```

(g) A learned typed PDDL operator for the Open-Chest skill. The predicate underlined in red indicates a problem-specific symbol that must be relearned for each new task, while the rest of the operator can be safely transferred.

Figure 13: Our approach learns that, in order to open a chest, the agent must be standing in front of a chest (symbol_13), the chest must be closed (symbol_4), the inventory must contain a clock (symbol_55) and the agent must be standing at a certain location (psymbol_8). The result is that the agent finds itself in front of an open chest (symbol_58) and the chest is open (symbol_59). type0 refers to the "agent" type, type6 the "chest" type and type9 the "inventory" type.

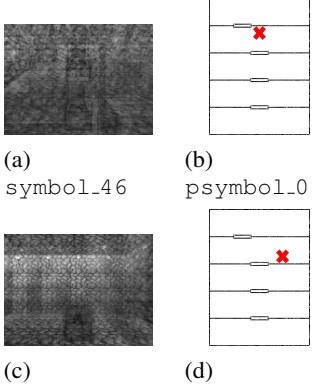

(a)
symbol_46

(b)
psymbol_0

(c)
symbol_11

(d)
psymbol_1

```
(:action Walk-to-partition-0-2a
 :parameters (?w - type0)
 :precondition (and (notfailed) (symbol_46 ?w)
          (psymbol_0))
 :effect (and (symbol_11 ?w) (psymbol_1)
       (not (symbol_46 ?w)) (not (psymbol_0)))
)
```

(e) Typed PDDL operator for a partition of the Walk-To option. The predicate underlined in red indicates a problem-specific symbol that must be relearned for each new task, while the rest of the operator can be safely transferred.

Figure 14: Abstract operator that models the agent walking to the crafting table. In order to do so, the agent must be standing in the middle of a room (symbol_46) at a particular location (psymbol_0). As a result, the agent finds itself in front of the crafting table (symbol_1) at a particular location (psymbol_1).

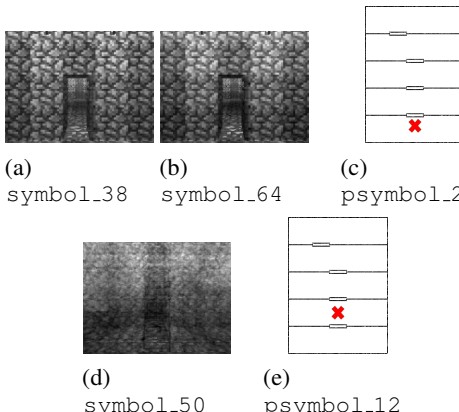

(a)
symbol_38

(b)
symbol_64

(c)
psymbol_24

(d)
symbol_50

(e)
psymbol_12

```
(:action Through-Door-partition-3-207a
 :parameters (?w - type0 ?x - type1)
 :precondition (and (notfailed)
         (symbol_38 ?w) (symbol_64 ?x)
         (= (id ?x) 1) (psymbol_24))
 :effect (and (symbol_50 ?w)
         (not (symbol_38 ?w)) (psymbol_12)
         (not (psymbol_24)))
)
```

(f) Typed PDDL operator for a partition of the Through-Door option. The predicate underlined in red indicates a problem-specific symbol that must be relearned for each new task, while the rest of the operator can be safely transferred.

Figure 15: Abstract operator that models the agent walking through a door. In order to do so, the agent must be standing in front of an open door (symbol_38) at a particular location (psymbol_24), and the door must be open (symbol_64). As a result, the agent finds itself in the middle of a room (symbol_50) at a particular location (psymbol_12).

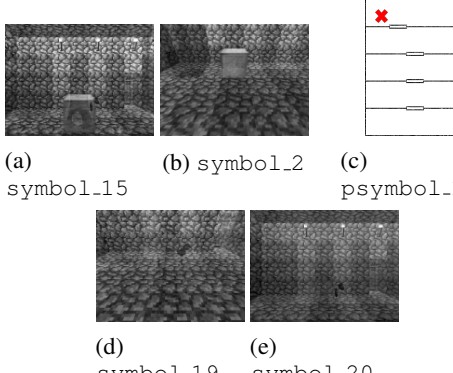

(a)
symbol_15

(b) symbol_2

(c)
psymbol_17

(d)
symbol_19

(e)
symbol_20

```
(:action Attack-partition-0-76a
 :parameters (?w - type0 ?x - type7)
 :precondition (and (notfailed)
     (symbol_15 ?w) (symbol_2 ?x)
     (psymbol_17))
 :effect (and (symbol_19 ?x) (symbol_20 ?w)
         (not (symbol_2 ?x))
         (not (symbol_15 ?w)))
)
```

(f) Typed PDDL operator for a partition of the Attack option. The predicate underlined in red indicates a problem-specific symbol that must be relearned for each new task, while the rest of the operator can be safely transferred.

Figure 16: Abstract operator that models the agent attacking an object. In order to do so, the agent must be standing in front of a gold block (symbol_15) at a particular location (psymbol_17), and the gold block must be whole (symbol_2). As a result, the agent finds itself in front of a disintegrated block (symbol_20), and the gold block is disintegrated (symbol_19).

# H EXAMPLES OF FAILURE CASES

Below are some examples of errors that occur when constructing our abstract representation. Since there are several phases involving clustering, classification and density estimation, we can expect various learning errors to occur throughout. These errors could have numerous causes, such as insufficient data or suboptimal hyperparameters.

## H.1 PARTITIONING ERRORS

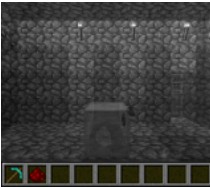
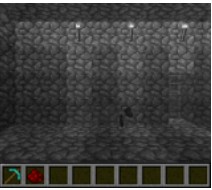

(a) Set of start states for one partition of the `Attack` option.

(b) Set of end states for one partition of the `Attack` option.

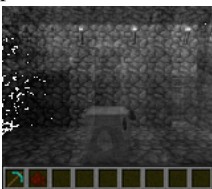
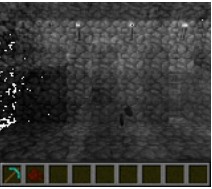

(c) Set of start states for another partition of the `Attack` option.

(d) Set of end states for another partition of the `Attack` option.

Figure 17: In the above example, the partitioning procedure has generated two partitioned options for breaking the gold block, where there should only be one. They are functionally equivalent, but because of the strange shadows on the left of the image patch and the subsequent PCA representation, the clustering algorithm has produced one extra partition.

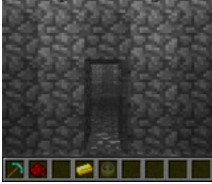
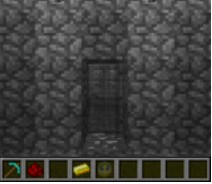

(a) Set of start states for one partition of the `ToggleDoor` option.

(b) Set of end states for one partition of the `ToggleDoor` option.

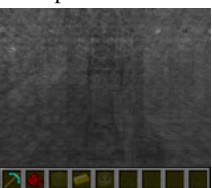
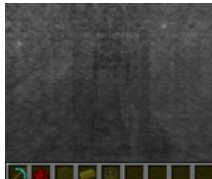

(c) Set of start states for another partition of the `ToggleDoor` option.

(d) Set of end states for another partition of the `ToggleDoor` option.

Figure 18: In this example, the partitioning has clustered noisy samples into an additional partition of the `ToggleDoor` option. While the top row shows the case where the state of the door changes from open to closed, the bottom row is a relatively useless noisy operator. We will subsequently learn a precondition and effect for this partition, but it likely will not be used by the planner.

## H.2 Precondition/Feature Selection Errors

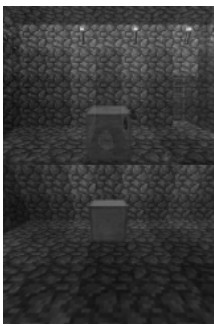

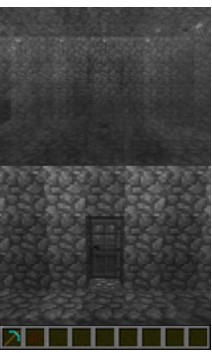

(a) The precondition for attacking the gold block. The top image represents the agent's view (in front of the block) while the bottom image is the state of the block (unbroken).

(b) The precondition for walking to a closed door. The top image represents the agent's view (in a room) while the bottom image is the state of the door (closed) and the state of the inventory.

Figure 19: In the left example, the classifier predicts that the gold block can be broken when the agent is in front of it. However, this is not quite correct, since the agent must also have the pickaxe to break the block. In this case, the issue occurs because the data only included states where the agent reached the gold block with the pickaxe. Therefore, the agent did not observe states where it was in front of the block without the pickaxe, and thus concluded that the pickaxe is irrelevant to the precondition. In the right example, the classifier has overfitted to the data and predicts that the agent can only walk to the door when it has the pickaxe.

## H.3 PPDDL Construction Error

The quality of the PPDDL operators depends on how accurately the precondition classifiers and effect estimators are learned. Any error in learning can result in imperfect PPDDL operators, as seen below.

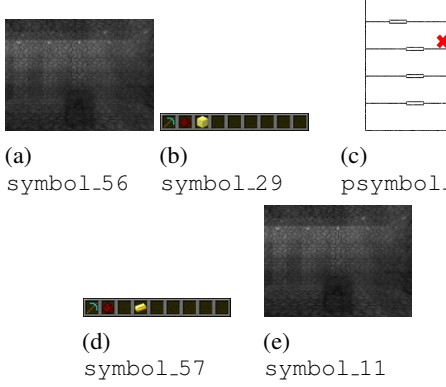

(a)
symbol_56

(b)
symbol_29

(c)
psymbol_1

(d)
symbol_57

(e)
symbol_11

```
(:action Craft-partition-1-240a
 :parameters (?w - type0 ?x - type9)
 :precondition (and (notfailed)
            (symbol_56 ?w) (symbol_29 ?x)
            (psymbol_1))
 :effect (probabilistic 0.21
         (not (notfailed))
                    0.79
         (and (symbol_57 ?x) (symbol_11 ?w)
         (not (symbol_29 ?x))
         (not (symbol_56 ?w))))
)
```

(f) Typed PPDDL operator for a partition of the Craft option.

Figure 20: Abstract operator that models the agent crafting a gold ingot. In order to do so, the agent must be standing in front of the crafting table (symbol_56) at a particular location (psymbol_1), and must have the gold block in its inventory (symbol_29). As a result, the agent finds itself in front of the crafting table (symbol_11), and now has a gold ingot in its inventory (symbol_57). This option is deterministic; however, due to estimation errors, the PPDDL operator predicts that it will only succeed with probability 0.79.

## H.4 Type Inference Error

We observe that occasionally the procedure will not discover the correct types. In the example below, instead of discovering a single type for all four doors, our approach predicts that one door is different from the others and is placed in its own class

Table 2: A grouping of objects into types. Note that one of the doors is allocated its own type.

| Type | Name | Object(s) |
|------|----------------|-----------|
| 0 | Agent | 0 |
| 1 | Pickaxe | 1 |
| 2 | Door1 | 2, 3, 4 |
| 3 | Door2 | 5 |
| 4 | Redstone Block | 6 |
| 5 | Gold Block | 7 |
| 6 | Chest | 8 |
| 7 | Inventory | 9 |

