# OpenReview forum: "Autonomous Learning of Object-Centric Abstractions for High-Level Planning"
_ICLR.cc/2021/Conference — Reject_

### Official Review · AnonReviewer4 · 2020-10-28
**Interesting idea; a few concerns.**

**Rating:** 4
**Confidence:** 3

**Review:**

The authors propose a planning approach that generalises across objects. Their approach groups objects into object types; if two objects have the same outcomes in planning then they are considered to be the same object. This is certainly a very interesting idea however I have some concerns about how this approach would work in more complex domains and how well the approach would generalise to imperfect object individuation.

Quality:

The partitioning of initial states I_o  (for each option) is achieved by “clustering state transition samples based on terminating states”. Since the state is defined by a set {f_a, f_1, …., f_n} I have a few questions: (1) How do you ensure that the clustering is invariant to the order of the objects, f_i? (2) The clustering must depend heavily on how the state is represented?

When learning the precondition model, if each object is treated independently, did the authors consider how this might be generalised to take into account interactions between objects? Similarly, when learning the effect model?

In this work the authors assume that the objects have already been individuated. While this is an important first step, it’s not clear how the approach would generalise if the object were imperfectly individuated. For example if only half of the door was visible in a particular frame? How would this approach scale to more complex observations?

Were there any fail cases when partitioning options based on their terminal state? Particularly in the Minecraft setting?

It would be good to show examples of which object types were learned  and if there were any fail cases.

Figure 6a is a very nice result — is it for BoxWorld of Minecraft? (Similar question for 6b).


Clarity:
The authors could make their contributions more clear in the introduction.

From section 2 Is not clear what {f_a, f_1, …, f_n} are in the authors’ implementation.  The authors say that they are pixels, but go on to say that their “state space representation assumes that individual objects have already been factored into their constituent low-level attributes”. The first might suggest that f_i are image patches of objects while the latter suggests that they are vectors representing object attributes, it would help to clarify.

The authors use of the term representation is not well defined. From “a sound and complete abstract representation must necessarily be able to estimate the set of initiating and terminating states for each option” the authors’ definition of the term representation is not clear.

At the end of the “Partitioned Options” section, it would be good to clarify which approach will be used in this paper.

Figure 1 is not very clear.

The feature selection process for determining which objects are are relevant to the precondition is only explained in the Appendix and may not be general. It is also not clear why this is needed? Why can the classifier not learn for itself which objects are useful and which to ignore? Is the implementation invariant to the ordering of the objects {f_i} for multiple i's?

When learning the precondition model do you treat each object independently and predict Pre(o) for each object? It would be good to clarify this early on.

It would help if in the introduction and experiments sections the authors were more clear about what they want to measure to show that their approach is beneficial in task transfer.



Originality and Significance:

The authors’ novel contribution is to propose clustering objects based on their effect, allowing transfer between tasks. This is an interesting way to group objects and could enable future research towards using objects in RL, for example by relaxing assumptions that the objects have already been individuated.


Typos:

In the introduction, PPDDL, is not defined.

“initial and terminating states I_o and \beta_o” -- \beta_o is the probability of a terminating state, not the state itself.

In section 3.1 and Figure 1, steps 2: It’s not clear that you *generate* a positional forward model.

“ to compute whether the skill can be executed there” —> do the authors mean option rather than skill here?

Figure 4 is referenced before Figure 2.

Check the caption on Figure 4.

---

> ### Author Response · Authors · 2020-11-19
> **Response (1/2)**
>
> Thank you for your detailed feedback, which we have incorporated into an update of the paper. To answer your questions:
>
> > How do you ensure that the clustering is invariant to the order of the objects, f_i?
>
> When clustering, we follow Konidaris et al (2018) and only use the data for objects whose values have been changed by the option (the so-called mask). Thus if an option changes object 1 and 3’s state variables, we ignore the other objects and only use the data from objects 1 and 3 when running DBSCAN. The ordering of the unaffected objects is thus irrelevant. However, the ordering does matter for the affected objects initially. Thus we may produce multiple partitions that in effect do the same thing (but operate on different objects). These are then collapsed into one if we determine those objects are of the same type.
>
> > The clustering must depend heavily on how the state is represented.
>
> Yes, this is indeed the case. DBSCAN, for instance, requires an epsilon hyperparameter that governs whether two states are considered to be in a neighbourhood of one another. Depending on the scale of the representation chosen, this epsilon would vary (e.g. if we used a pixel-based representation with values [0-255], then epsilon would be higher than if the values were between [0, 1] ).
>
> > When learning the precondition model, if each object is treated independently, did the authors consider how this might be generalised to take into account interactions between objects? Similarly, when learning the effect model?
>
> Yes, this is a good point. We do account for this. Just before we learn the precondition, a feature selection step determines which objects are most important in correctly classifying positive/negative states.  An example of this in action can be seen in Figure 9 in the appendix: opening the chest requires three objects: the agent, the chest and the inventory. All three of these objects are identified as being important for the precondition (the chest cannot be opened without the clock). Effects involving multiple objects are also accounted for; however, these are easier to compute because we can use the change in state to determine which objects changed. For example, picking up a gold block affects the agent’s view (since it cannot see a block anymore), the state of the gold block (since it is not present) and the state of the inventory (which now has a gold block)
>
> > In this work the authors assume that the objects have already been individuated. While this is an important first step, it’s not clear how the approach would generalise if the object were imperfectly individuated. For example, if only half of the door was visible in a particular frame? How would this approach scale to more complex observations?
>
> This is a great question. We did not account for this to keep the experiments as clean as possible. However, the effect of this would be the following: a) we would produce more predicates that semantically describe the same thing but whose representations are significantly different. This would require more  PDDL operators to model these cases. For example, we may end up modelling a door opening as a probabilistic effect where X% of the time, the door is open and fully in frame, and (1-X)% of the time the door is open, but only half in the frame. Then, walking through an open door would need to have preconditions for the door being open (and completely in frame) as well as the door being open (and half in frame). Of course, if we have an appropriate similarity check, we can potentially overcome this if we deduce that the door in frame and out of frame look similar enough that they can just be modelled by a single predicate instead of two. However, if they are just too different, then we would end up producing an extra N predicates (one for each sufficiently different view) along with operators that include each of the N predicates. Imperfect individuation could also result in a POMDP (we only handle the MDP case here). We could potentially overcome this by “completing” the rest of the state of the door from memory, or having a distribution over it.
>
> > Were there any fail cases when partitioning options based on their terminal state? Particularly in the Minecraft setting?
>
> Yes, there were indeed errors in this process. For example, because there are 4 doors in the task, we would expect 8 partitions of the ToggleDoor option (4 for opening and 4 for closing). Instead, we produced 10 partitions which included an extra opening/closing case. When we visualise these extra cases, the image produced is just noise. We therefore learn extra PDDL operators that are completely extraneous (and don’t actually occur in practice). We have added multiple examples of failure cases to the appendix.

---

> > ### Author Response · Authors · 2020-11-19
> > **Response (2/2)**
> >
> > > It would be good to show examples of which object types were learned and if there were any fail cases.
> >
> > We have also added this to the appendix. The one obvious failure was that instead of producing a single “door” object that included all 4 doors, the procedure produced two door classes. The first 3 doors belonged to door class 1, but the fourth door was determined to belong to door class 2.
> >
> > > Figure 6a is a very nice result — is it for BoxWorld of Minecraft? (Similar question for 6b).
> >
> > Both figures are for Minecraft. We used the BoxWorld domain only for explanatory purposes.
> >
> > > From section 2 Is not clear what {f_a, f_1, …, f_n} are in the authors’ implementation. The authors say that they are pixels, but go on to say that their “state space representation assumes that individual objects have already been factored into their constituent low-level attributes”. The first might suggest that f_i are image patches of objects while the latter suggests that they are vectors representing object attributes, it would help to clarify.
> >
> > We will clarify that we indeed mean image patches in the case of Minecraft (although in practice we work with the PCA weights to make clustering and density estimation easier). In general though, we imagine these low-level attributes can be anything, be it pixels, voxels or a flat vector representation. The important takeaway is that the representation for an individual object is known.
> >
> > > The authors use of the term representation is not well defined. From “a sound and complete abstract representation must necessarily be able to estimate the set of initiating and terminating states for each option” the authors’ definition of the term representation is not clear.
> >
> > We will clarify that by representation we mean that we wish to represent our environment using a learned state and action abstraction. This could take the form of an abstract MDP, with abstract states and actions, or a PDDL representation with PDDL operators and predicates (which amounts to the same thing).
> >
> > > The feature selection process for determining which objects are are relevant to the precondition is only explained in the Appendix and may not be general. It is also not clear why this is needed? Why can the classifier not learn for itself which objects are useful and which to ignore? Is the implementation invariant to the ordering of the objects {f_i} for multiple i's?
> >
> > It may indeed be the case that the SVM itself can learn to ignore irrelevant objects. However, without an explicit feature selection step, we would produce PDDL operators where every single object is in the precondition. The precondition would need to be a massive disjunction to describe a particular operator, effectively saying that for the objects that matter, they must be in particular states, but for all other objects, they can be in *any* state. We would need to explicitly enumerate all the possible states the irrelevant objects could be in to accomplish this. Our crude feature selection process is invariant to the ordering of objects, since we simply loop through each in turn and check if it increases classification accuracy. If it does, then it is added to the precondition.
> >
> > > When learning the precondition model do you treat each object independently and predict Pre(o) for each object? It would be good to clarify this early on.
> >
> > We will clarify that for learning the preconditions we do not treat objects independently. Once the feature selection step has identified the objects of importance, we learn a single classifier over all the objects jointly.
> >
> > Thanks for identifying other clarity issues and typos which we will correct.

---

### Official Review · AnonReviewer3 · 2020-10-28
**Paper proposes a way to create object centric representation that can transferred across task with same objects**

**Rating:** 5
**Confidence:** 1

**Review:**

- No Baseline Comparisons: Paper proposed a way to generate object level representation that can be used across the tasks with same objects. Authors claim that this should reduce the number of environment interactions required to solve the new task. However, there is no baseline comparison being done to figure out how sample efficient it is.

- Paper assumed that tasks are solvable from set of provided options, will the method work of the task is out of provided options space ?

---

> ### Author Response · Authors · 2020-11-19
> **Response**
>
> Thank you for your feedback.
>
>
> > No Baseline Comparisons: Paper proposed a way to generate object level representation that can be used across the tasks with same objects. Authors claim that this should reduce the number of environment interactions required to solve the new task. However, there is no baseline comparison being done to figure out how sample efficient it is.
>
>
> We are unaware of any existing methods that would be appropriate as a baseline comparison. RL methods that learn a value function or policy do not support the ability to plan to multiple different goals after learning. Model-based methods would be more appropriate, but they do not learn an explicit PDDL representation that can be used by task-level planners. It is also unclear how to apply any of the above to tasks whose state space varies (with different numbers of objects).  However, our results do show that the abstractions are indeed transferable between tasks and that there is benefit to doing so (as opposed to learning from scratch, which we believe to be the best available baseline for this task).
>
> > Paper assumed that tasks are solvable from set of provided options, will the method work of the task is out of provided options space ?
>
>
> No, unfortunately not. Our approach builds a state abstraction based on the preconditions and effects of an agent’s skills. If the agent does not have skills that are sufficient to solve a particular goal, then naturally we cannot build a state abstraction to solve the task (since the agent does not possess actions that could solve the task in the first place).

---

### Official Review · AnonReviewer1 · 2020-10-29
**The paper is too high-level and hard to digest.**

**Rating:** 4
**Confidence:** 4

**Review:**

The paper proposes an approach to learn a Probabilistic PDDL representation for tasks and generalize to new tasks with very small number of additional samples. It does that by first learning a compact and lifted representation across different training tasks that get the gist of multiple different objects, then converts it into symbolic representation (in PDDL) for future planning.

I think the problem itself is super interesting and important: how to combine neural based and traditional approach remains a key challenge for AI community. And this paper seems to make a good attempt towards addressing it.

However, I found this paper is hard to follow without any detailed descriptions of the actual algorithms.  The description is mainly high-level, many symbols/terms are mentioned but not defined, and there is no clue how to implement the idea. There seems to be 5 steps (Sec. 3.1-3.3 and Fig. 1). But it is not clear at all how the entire system work. Here I only list a few:

1. what are the features defined in Sec. 4.1?
2. Are the options o learned or pre-defined?
3. How is the clustering /precondition classifier/feature selection in Sec 3.1 done?
4. How are the profiles in Def. 1 computed?
5. What is symbol_10 in Fig. 2? How is it learned?
6. How does the algorithm replace three propositions AOnTable, BOnTable and COnTable with a single predicate InHand? This is the “lifting operation” mentioned in the paper but I didn’t find any algorithmic description.

While the author defers “the exact implementation and domain details to the appendix”, I don’t find many details unfortunately, except for the learned/generated results.  The algorithm block in appendix F doesn't help much, with many undefined subroutines and symbols.

Given the difficulty in understanding the details, I feel that the paper is a bit premature for top-tier conference.

======

After reading the author's comments I still keep the score.

After rebuttal/revision, the paper still has a lot of steps, many of them are human designed and are not well-defined. For example, in the author response, they said "their effect distributions look similar, and so are merged into one class type", what is the criterion to merge them together? In the revised paper, what is the procedure "COMPUTEMASK" defined?  In line 38 of the algorithm proposed in Appendix F in the revised paper, what does the sign of "approximate equal" mean?  Overall, this makes it hard to reproduce and it is not clear whether the proposed approaches can be applied to other problems than the specific tasks mentioned in the paper.

---

> ### Author Response · Authors · 2020-11-19
> **Response**
>
> Thank you for the feedback! We apologise for the lack of clarity. While we described the details in Appendix E, we see that our writing there has failed to communicate those details, so we'll work hard to rewrite that section so that it is clearer. We have added pseudocode in the appendix with additional subroutines to hopefully address some of these issues. The code is thousands of lines long, and so capturing every minute detail in the pseudocode is infeasible.
>
> Please let us know if there are any sections that are still unclear, and we will work to clarify them.
>
> To answer your specific questions:
>
> > what are the features defined in Sec. 4.1?
>
> As in Section 4.1, each block is represented by a vector of length 2. The first element represents what is above the block (one of three options - empty space, another block, a table) and the second element represents what is below it (again, with the same three options). The hand is simply a single boolean feature (is there an object in the hand?)
>
> > Are the options o learned or pre-defined?
>
> In this work, the options are pre-defined, but they could also be learned using an existing option-discovery method. Different options would lead to different representations. We choose to specify them here upfront so that we can better understand the learned abstractions, without conflating our results with the option-learning method.
>
> > How is the clustering /precondition classifier/feature selection in Sec 3.1 done?
>
> The same approach is applied to both the Blocks World and Minecraft. To expand on Appendix E, after collecting data, we first use DBSCAN to cluster the end states of each option. Clustering is done only on state variables whose values have changed (e.g. if an option is executed and the state of a door does not change between the start and end states, then that door’s variables aren’t used in the clustering procedure). This produces partitioned options which approximately preserve the subgoal condition. Then, for each of these partitioned options consisting of start/end-state data, we fit an SVM classifier. Before this, we implement a crude feature selection procedure to determine which objects are most important for the classifier. We do this incrementally by adding objects one at a time and refitting the classifier. If the score improves, we keep the object. At the end of this procedure, we fit the classifier to the selected objects.
>
> > How are the profiles in Def. 1 computed?
>
> The profiles in Def 1 are essentially the output of the first two steps (and the output of Konidaris 2018). At this point, we have learned a PDDL representation for all the partitioned options, which consists of preconditions and effects. The preconditions and effects are all represented as propositions - distributions over low-level states. Thus to compute the profiles here, we just need to extract, from each PDDL operator, only the propositions which refer to the object in question. All other propositions which are distributions over other objects) are removed.
>
> > What is symbol_10 in Fig. 2? How is it learned?
>
> As in the caption, symbol_10 represents the distribution over states where the hand is empty and all the other objects can be in any state. Thus in our visualisation, the hand is always empty, but the other blocks can occupy any other position (since the distribution does not refer to them).
>
> > How does the algorithm replace three propositions AOnTable, BOnTable and COnTable with a single predicate InHand? This is the “lifting operation” mentioned in the paper but I didn’t find any algorithmic description.
>
> Apologies for the typo. That sentence should read “...the agent can replace them all with a single predicate **OnTable(X)**...” To achieve this, the agent looks at the computed PDDL effects for all options. For the partitioned options that affect each block, their effect distributions look similar, and so are merged into one class type. Because there are 3 blocks, all the propositions and operators that refer to two of the blocks are removed, and the remaining propositions and operators are converted into  parameterised predicates and operators.

---

### Official Review · AnonReviewer2 · 2020-10-30
**The paper has a few issues**

**Rating:** 3
**Confidence:** 4

**Review:**

The paper presents an approach for object-centric representation learning for planning to accomplish complex tasks. The learned representations are at an abstract level, resulting in desirable knowledge transfer capabilities between tasks. The learned action knowledge is represented using PDDL. Experiments have been conducted using blocks world and Minecraft domains. Results show that the agent was able to learn useful operators (actions) and that learned actions can be applied to different tasks.

The object-centric idea is highlighted in the paper, though the work is more about learning symbols for abstraction, which is not new. The issue is that many of the "learning abstraction" works have been demonstrated in much more complex domains. For instance, the work of Konidaris et al 2018 (cited in the paper) has enabled robots to learn action preconditions and effects in the real world. In comparison, this work does not go beyond toy problems.

The developed approach is more like an integration of a few existing methods, and the connections among the pieces are rather weak (see Figure 1). For instance, once the actions are learned, it looks like the agent faces a planning problem, and the planner does not have a way to go back to improve its learned representation. Since each component introduce its own errors, there's the cumulative error that can potentially make the whole system rather unstable. Some co-learning functionalities will be good for future work.

The experiment section is relatively weak. For instance, the blocks world domain was mostly used as a demonstration platform. There were a couple of examples presented, while there were no statistical results discussed in the paper. Figure 6 is on the Minecraft domain, where the baseline (such as no transfer) is very weak. The results are not convincing to support the claims on transferability and learning efficiency.

-------

The reviewer appreciates very much, and has read the response letter, which is mostly about clarity though it's not the main concern of the reviewer.

---

> ### Author Response · Authors · 2020-11-19
> **Response**
>
> Thank you for your review and feedback.
>
> > The developed approach is more like an integration of a few existing methods, and the connections among the pieces are rather weak (see Figure 1).
>
> Our proposed method is a new approach that extends Konidaris et al (2018), and so it contains a lot of machinery from that work. However, the object-centric lifting process is new and, as we show, that results in the ability to transfer between domains, the inability to do so being a major failure case of the prior work.
>
> > In comparison, this work does not go beyond toy problems.
>
> Although it is a software-based environment (we unfortunately do not have access to a physical robot) we disagree that the Minecraft domain is a “toy domain”. While prior work has been applied to a real robot, it was in a relatively self-contained and constrained environment. Our Minecraft setup is actually much larger, and has higher-dimensional inputs. Additionally, the solution trajectory here is significantly longer than the robot domain. In addition to this, the Malmo platform is extremely challenging for a number of reasons:
>
> 1. Interacting with the Malmo platform is very slow which means that, much like the real world, we cannot collect hundreds of thousands of samples ([1] shows that MineRL, a much faster fork of the Malmo platform, is 15x slower than Atari).
> 2. Because communication with the platform is asynchronous, the domain is non-deterministic.
>
> All of these issues make Malmo a tougher challenge than existing “harder” domains like Atari.  This is one of the reasons that a lot of work in Minecraft has previously focused on relatively simple tasks like navigating and walking to blocks [2, 3]. Tasks that require long term planning, such as our procedurally generated ones here, are still far out of reach. This can be seen in the MineRL competition [4], which requires such long-term planning. Current approaches completely fail at these kinds of tasks (there were more than 500 entries from around the world), even with millions of human demonstrations.
>
> > Figure 6 is on the Minecraft domain, where the baseline (such as no transfer) is very weak
>
> In terms of baseline comparisons, we are unaware of existing methods that would be appropriate.. RL methods that learn a value function or policy do not support the ability to plan to multiple different goals after learning. Model-based methods would be more appropriate, but they do not learn an explicit PDDL representation that can be used by task-level planners. Thus, beyond the work of Konidaris et al (2018) - which is equivalent to the “no transfer baseline”, there seems to be no other appropriate candidate for comparison.  It is also unclear how to apply existing approaches to tasks whose state space varies (with different numbers of objects).  However, our results do show that the abstractions are indeed transferable between tasks and that there is benefit to doing so (as opposed to learning from scratch, which we believe to be the best available baseline for this task).
>
> [1] Küttler, Heinrich, et al. "The NetHack learning environment." Advances in Neural Information Processing Systems 33 (2020).
> [2] Tessler, Chen, et al. "A deep hierarchical approach to lifelong learning in Minecraft." Proceedings of the Thirty-First AAAI Conference on Artificial Intelligence. 2017.
> [3] Shu, Tianmin, Caiming Xiong, and Richard Socher. "Hierarchical and Interpretable Skill Acquisition in Multi-task Reinforcement Learning." International Conference on Learning Representations. 2018.
> [4] Guss, William H., et al. "MineRL: A Large-Scale Dataset of Minecraft Demonstrations." International Joint Conference on Artificial Intelligence (2019).

---

### Decision · Program_Chairs · 2021-01-07
**Final Decision**

**Decision:**

Reject

**Comment:**


The paper proposes a general framework for learn object-centric abstractions represented using PPDDL (a probabilistic planning language).  The work assumes that objects and their attributes / features are identified.  The key contribution of the paper appears to be proposing to group individual objects into object types based on whether objects have the same outcome in planning. Using the learned object types, it would then be possible to transfer learned operators from one task to another.  The framework is demonstrated on block world (blocks are stacked on top of on another) and minecraft.


Review Summary: Initially, the submission received negative to borderline reviews with R4 being the most positive (score 6), and R1, R2, R3 being more negative (scores 4, 3, 5).  After discussion, R4 lowered their score to 4 and indicated that they felt the work was not ready for acceptance at ICLR.  Overall, there was limited discussion by the reviewers.  Reviewers (R2,R4) found the direction of the work promising and interesting.  After the author response, reviewers indicated that the revision and author response clarified some points, but believe that the work is not yet ready for acceptance, as 1) there is a significant amount of hand-crafting required and 2) parts of the approach is still not clearly specified.

Clarity: As some reviewers note, the description of the framework is at a very high level, making it difficult to follow with missing details on specific details of how the object types are groups.  The specific aspect of the work that is novel is also not clearly stated, thus making it difficult to judge the originality and significance of the work.  After revision (the authors added brief paraphrase to the introduction to clarify the contribution, and additions to the appendices providing more details on how the difference steps work for the Minecraft scenario as well as failure cases), the manuscript is improved but the overall manuscript is still difficult to follow.

Pros:
- Interesting and important problem (combining probabilistic/neural approaches with symbolic approaches) that is timely and deserves attention
- The idea of clustering objects based on their effect is interesting (R4).
- The framework proposed by the paper is interesting and potentially useful direction and can stimulate followup work

Cons:
- The paper is difficult to follow with symbols/terms that are not clearly defined and missing details. (R1) The specific contributions of the work, wrt prior work, is also not clearly stated.
- The novelty/contribution of the work on top of existing work (Konidaris et al 2018, Ugur & Piater 2015, etc) is not that clear (R2)
- The experimental setup is weak with limited comparisons and no statistical results. Overall, reviewers felt the results are not convincing enough to support claims on transferability and learning efficiency.
- Lack of baselines comparisons (R3).  In the rebuttal, the authors argue that there is no appropriate baselines.
- The set of steps that is involved is fairly complex (R1), with many important details provided in the appendix
- There are concerns about the generalization of the approach as many of the steps are handcrafted (R1, R4).  In the provided scenarios, many of the steps, including the set of provided options, and representation of objects, are manually designed.

Recommendation:
The AC agrees with the reviewers that the paper is not ready for acceptance to ICLR.  It is the AC's opinion that the work addressing a very interesting problem and
would potentially be of interest to the community.  However, the exposition of the paper needs to be improved so that 1) the contribution of the work over prior work (Konidaris et al 2018, Ugur & Piater 2015, etc) is clearer 2) the assumptions and details of the proposed method is also clearer and easier to follow.  The authors are encouraged to improve their work and resubmit to an appropriate venue.